# Alignment of Wikidata lexemes and Det Centrale Ordregister

**Finn Årup Nielsen**
DTU Compute
Technical University of Denmark
`faan@dtu.dk`

## Abstract

Two Danish open access lexicographic resources have appeared in recent years: lexemes in Wikidata and Det Centrale Ordregister (COR). The lexeme part of Wikidata describes words in different languages and COR associates an identifier with each different form of Danish lexemes. Here I described the current state of the linking Wikidata lexemes with COR and some of the problems encountered.

## 1 Wikidata

Wikidata (Vrandečić and Krötzsch, 2014), the large collaboratively edited knowledge graph, has a special namespace for words and the first lexeme in this namespace was created in 2018 (Nielsen, 2019).[1] Since then the number of lexemes has steadily grown. In February 2020, Wikidata had over 250,000 lexemes (Nielsen, 2020). As of January 2023, there were over 980,000 lexemes, over 12 million forms and over 290,000 senses. Wikidata records, e.g., over 213,000 German lexemes and over 16,000 Danish lexemes. A lexeme in Wikidata describes forms and senses with a growing set of properties. Each lexeme, form, and sense has a unique identifier. An example of a property for Danish lexemes is the link to words in the Danish wordnet resource DanNet (Pedersen et al., 2009).

Four templates for Lucas Werkmeister's *Wikidata Lexeme Forms* tool[2] are so far set up to help enter Danish words in Wikidata: For adjectives, verbs, and common and neuter gender nouns.

## 2 Det Centrale Ordregister

Det Centrale Ordregister (COR) (Dideriksen et al., 2022) is a Danish lexicographic resource with an initial test dataset released in May 2022 and since then updated. The resource and its description are available at `https://ordregister.dk/`. COR mostly describes the forms of lexemes and gives each form an identifier. Work on the semantic part of COR is underway (Nimb et al., 2022; Pedersen et al., 2022), but here I will not consider this part. The version of COR I consider is version 1.02 of the core COR and the 1.0 version for COR-EXT.

## 3 COR Wikidata properties

For linking Wikidata and COR, the Wikidata community has created two Wikidata properties for COR identifiers: One for the lexeme and one for the form. For instance, for the Wikidata lexeme *Sudan* (L993787) the COR lemma ID, level 1 identifier (P10831) is "COR.09978" while the COR form, level 1 identifiers (P10830) are "COR.09978.500.01" and "COR.09978.500.01" for the non-genitive (*Sudan*) and the genitive (*Sudans*) forms, respectively. The current regular expression contraint for the lexeme form is `COR\.(EXT\.\d{6}|\d{5})` allowing the core COR as well as the level 2 COR-EXT identifiers, — and it could be extended if other resources appear. The two COR Wikidata properties were created in June 2022. As of February over 2,100 COR lexemes/lemmas and over 1,800 COR forms are linked from Wikidata.[3] Around 390 Danish Wikidata lexemes have so far been annotated as not being present in COR. Most of these lexemes are compounds. Proper nouns and a few interjections comprise most of the rest.

---

[1] The Sumerian word for mother, `https://www.wikidata.org/w/index.php?title=Lexeme:L1&action=history&dir=prev`

[2] `https://lexeme-forms.toolforge.org`

[3] `https://ordia.toolforge.org/statistics/`

## 4 What is a lexeme and a form?

For linking Wikidata and COR it is important that there is a correspondence between the items of the two resources.

Danish words may have spelling variants, e.g., *højtaler* and *højttaler*. In COR, they are grouped under the same lexeme and as different variants. In Wikidata, spelling variants may be grouped under the same form. For instance, the English form L1347-F1 is currently listing both *color* and *colour*, separating them with different language specifications (*en* and *en-gb*). In Danish, the spelling variants do not arise due to different languages and the current Wikidata interface cannot handle spelling variants within the same language in one form. So far the Danish lexemes in Wikidata create separate forms for each spelling variants, e.g., L229388-F1 is *højtaler* and L229388-F2 is *højttaler*, making one COR spelling variant map to one Wikidata form. Two Wikidata spelling variant forms can be linked with the symmetric *alternative form* property (P8530).

In Wikidata, we have so far followed the scheme of *Den Danske Ordbog* (DDO) for Danish nouns with multiple genders and use only one lexeme for these cases. For instance, *øl* has common and neuter versions of the noun under the same lemma in DDO.[4] In Wikidata, this is also one lexeme: L39743. In COR there are two lexemes for *øl*: COR.45830 and COR.48125, thus in this case we get a one-to-two relationship between Wikidata and COR. Other examples of this type are *vand* and *kirsebær*. COR has *safran* (COR.93857) also as both common and neuter gender represented with one lexeme. The indefinite form, which does not reveal the gender, has two forms in COR: COR.93857.110.01 and COR.93857.120.01. Such a word often occurs in the bare form with no article or morphological gender suffix, so it may be impossible to detect the gender of the form in the context. Lexeme linking will have an ambiguity in this case. The current entry in Wikidata has just one (non-genitive) indefinite form. The same issue appears, e.g., for *kanel* (COR.57435) and *druk* (COR.86399).

Homographs that only have one gender are well-aligned between COR, Wikidata, and DDO, e.g., the noun(s) *fyr* has 3 separate lexemes in Wikidata and also has 3 separate lexemes (lem-

mas) in COR.

Superlative may be regarded as a derivation from the positive form (Hansen and Heltoft, 2019, p. 186–7), but both in COR and the Danish lexemes in Wikidata, the superlative is forms under normal adjective lexemes that also has the positive and comparative forms.

Centaur nouns (Danish: kentaurnominaler) are developed from verbs with an *-en* suffix. In both Wikidata and in COR (i.e., COR-EXT), they are regarded as separate lexemes, — and not a form of the verb lexeme.

Danish *perfektum participium* (adjective forms of the verb) exists in the borderland between being forms of a verb and an adjective derived from a verb (Holm and Christensen, 2019, p. 118). So should *perfektum participium* forms be grouped under a separate lexeme? COR does usually not record the adjective form of the verb separate from the verb. For instance, *barberet* in *et barberet ansigt* (a shaved face, COR.38323.213.01) is grouped under the verb *barbere* (COR.38323). There is an advantage in making a derived lexeme for the adjective forms, as it allows for the description of the sense, e.g., for *barberet* the antonym *ubarberet* can be specified. In Wikidata, the sense of the adjective *barberet* (L940943) is linked via the antonym property (P5974) to the sense of *ubarberet* (L940942) and vice versa. If the adjective was not a separate lexeme, but just part of the verb lexeme, it would not be straightforward to make this antonym link. With the adjective *barberet* as an individual lexeme, words such as *glatbarberet* and *nybarberet* becomes compounds, and the senses of the two compounds can be linked to the sense of *barberet* via the hypernym property.

In COR, it is not all verb-derived *-et* adjectives that are not separated from a corresponding verb, e.g., *snobbet* (COR.24113) is separate from the verb *snobbe* (COR.37973). In these cases, the verb still has the *perfektum participium* forms. Surprisingly, *overstimulere* is in COR as a verb (COR.32555) and *overstimuleret* is not in COR as a separate adjective, while *understimulere* is not in COR, while *understimuleret* is a separate adjective (COR.23107).

Adverbs from adjectives are grouped under the adjective lexeme in COR. In Wikidata, lexemes must be associated with a single lexical category (e.g., adverb or adjective). We have created separate lexemes for Danish adverbs from adjectives

---

[4] https://ordnet.dk/ddo/ordbog?query=%C3%B8l.

in Wikidata (for the few entered so far), e.g., the adverb *hurtigt* with forms *hurtig*, *hurtigere* and *hurtigst* has the Wikidata lexeme identifier (L691405) that is separate from the adjective lexeme (L42201) and with the COR lemma identifier (COR.15444) duplicated between the two Wikidata items.

## 5 Lexical categories

Below I will go through some of the most important lexical categories and how Wikidata and COR aligns. Small lexical categories such as conjunctions and prepositions are fully linked, with a few oddities: DDO and Wikidata have *dels* as a conjunction. It is an adverb in COR. Wikidata has *plus at* as a conjunction. It does not exist in COR.

### 5.1 Pronouns

What is a form and a lexeme for a pronoun varies between resources. For instance, COR has *han*, *ham* and *hans* (he, him, his) collected in one lexeme (COR.01880), while DDO separates them among three different dictionary entries. In Wikidata we have followed DDO and have three different lexemes for the three Danish words. In the English part of Wikidata, *he*, *him*, *his* and *himself* are collected into one lexeme (L485).

There are unusual grammatical features for some of the forms in COR, e.g., *ingens* has two forms indicated with "pron.gen" (pronoun, genitive) and "pron.sg.fk.gen" (pronoun, singular, common gender, genitive). Currently, the former links to the plural genitive form in Wikidata.

### 5.2 Numeral

COR has genitive forms for numerals, while the few numerals in COR-EXT do not. We started with no genitive forms for Danish numerals in Wikidata but now have begun to enter them. A few numerals in COR-EXT includes Arabic numerals, e.g., "94" as a form (COR.EXT.129099.600.02). Arabic numerals do not appear in the core COR and we have so far not included them in Wikidata.

### 5.3 Nouns

The non-genitive forms are not annotated as explicitly non-genitive in COR. In Wikidata, we can explicitly annotate forms that are not genitive with the non-genitive item (Q98946930).

Some forms are recorded in CORs but should perhaps not be present, while other forms are not recorded but perhaps should be. For instance, *drukning* (COR.61517) has no plural forms in COR while Den Danske Ordbog (DDO)[5] records plural forms and an Internet search[6] returns some examples of the plural form. *forundring* has neither plural forms in COR nor DDO but appears though infrequently, e.g., in the title "Syv forundringer over resiliensbegrebet". COR records *vejr* (COR.44355) with plural forms while the compound *blæsevejr* (COR.66305) is without plural forms. DDO records them without plural forms and in Wikidata *vejr* has been labeled a singulare tantum.

At one point Danish nouns in Wikidata did not record the genitive forms. This was based on a discussion on the Danish -*s* as a clitic.[7] The Danish genitive -*s* can attach to phrases, e.g., even adverbs (Herslund, 2001), so if nouns should have genitive why not other lexemes from other lexical categories? Given that COR is representing nouns with genitive forms, we have started to add genitive forms for nouns in Wikidata.

Centaur nouns are often missing from dictionaries (Rajnik, 2009; Gregersen, 2014; Hansen and Heltoft, 2019). Many centaur nouns are missing in the core part of COR, but are listed in the COR-EXT, e.g., *søgen* and *banden* are not in the core part, but in COR-EXT. *indsynken* and *indsætten* used in medical texts ("indsynken i sig selv" and "akut indsætten") are centaur nouns that are missing in both resources. Other missing centaur nouns are *malen* and *truen*, both described in (Holm and Christensen, 2019). Centaur nouns are claimed to have no genitive form (Hansen and Heltoft, 2019, p. 612). Nevertheless COR-EXT records genitive forms, e.g., *søgens* and *bandens* and the rare genitive forms appear: An Internet search yields "denne søgens forløsning" and "denne søgens neutralitet".

### 5.4 Verbs

The initial entries of Danish verbs in Wikidata did not model the passive forms completely: The -*es* forms were annotated as one passive form. Following COR, we have now started to annotate the Danish verbs in Wikidata with two -*es* forms: The

[5] https://ordnet.dk/ddo/ordbog?query=drukning&search=Den+Danske+Ordbog

[6] A Google search with ""drukninger" site:dk"

[7] See https://www.wikidata.org/wiki/Wikidata_talk:Lexicographical_data/Archive/2018/10

passive–infinitive and passiv–present tense.

Verbs with multiple different conjugation are entered in Wikidata as different forms, e.g., *fise* has *fes*, *fiste* and *fisede* with the same temporal and grammatical features. Further Danish verbs that diverge from the normal 9-form conjugation scheme in Wikidata are deponent verbs that lack some forms and verbs ending with *-ere* where the imperative has an alternative spelling.

In COR 1.02, there are unusual *perfektum participium* forms for some of the common verbs, e.g., *hafte* (COR.30035.214.01) and *skullede* (COR.30128.214.01). They are not found in Retskrivningsordbogen nor in DDO.

## 5.5 Adjective

It has been unclear which grammatical features should be assigned to the different forms of Danish superlative (*-st* and *-ste*). Standard works in Danish grammar regard them as a kind of definite (or definite-ish) inflection (Diderichsen, 1962; Hansen and Heltoft, 2019). Danish Wiktionary[8] and the Swedish lexemes in Wikidata use the grammatical features predicative and attributive, see, e.g., *rolig* (L53287). COR 1.02 has 3 superlative forms, e.g., for *travl*: *travlest* (COR.15021.305.01, singular, indefinite), *travleste* (COR.15021.306.01, singular, definite) and *travleste* (COR.15021.307.01, plural). In Wikidata lexemes, we have only recorded two superlative forms: indefinite (*-st*) and definite (*-ste*).

COR includes comparative and superlative forms of adjectives that er quite rare, e.g., *radioaktivere*, *radioaktivest* and *radioaktiveste* from COR.26147 and *ugennemtænkere*, *ugennemtænktest*, *ugennemtænkteste* from COR.26148. With a simple Internet search, I was not able to find any examples of these forms, other than electronic dictionaries, while the periphrasic versions (e.g., *mere radioaktiv*) occur. Even *apropos* and *forleden*—which in the online version of Retskrivningsordbogen are regarded as uninflectable adjectives[9]—have comparative and superlative forms in COR 1.02.

Some of the superlative forms in COR are questionable: Adjectives that have no positive form, e.g., *ypperst* is specified with the positive form *ypperst*, but this should rather be superlative. COR

has highly unusual *ypperstest* as the superlative form. Special cases are the compas adjectives *østre*, *nordre*, *vestre* og *søndre*. According to DDO they are originally comparative to *øst*, *nord*, *vest* og *syd*. COR regards these adjectives as in their positive form and records a comparative form, e.g., *nordrere*, and the superlative forms.

Nominalization of an adjective may result in a new lexeme noun in COR, e.g., the noun *indre* (COR.48793) is separate from the adjective and has a genitive form. These nominalizations are rare and adjectives do not have genitive in COR. Genitive forms have so far not been added to the adjectives in Wikidata.

## 6 Semantics

A few of the entries in COR have a short text for disambiguation of homographs. In a few cases it has been used as a gloss to the sense of a Wikidata lexeme, e.g., for *æg* COR disambiguates with "fx: fugleæg" (e.g., bird's egg) and "skarp kant" (sharp edge) for the two homographs and the Wikidata sense L39239-S1 "skarp kant" has been noted as a gloss referencing COR. For the other homograph, Wikidata has currently two senses (biological egg and egg as food) making the application of the disambiguation text as a sense gloss difficult. Such a case is not uncommon making automated setup or alignment of senses based the COR disambiguation text not feasible or at least difficult.

## 7 Discussion

There are Wikidata tools for mass-entry of lexemes and with COR data Danish Wikidata lexemes could be set up en masse. So far I have setup the links manually exploring the problems of ontology linking the two resources. I find *perfektum participium*, inflections of adjectives and nouns with both neuter and common gender are among the issues where one should be careful with matching. After the publication of COR, we have changed the entry of genitive for nouns and numerals and passive forms of verbs in Wikidata. I suspect that we might see a revision of inflections of adjectives in COR around comparative and superlative forms.

## Acknowledgments

Thanks to Peter Juel Henrichsen and Thomas Widmann for discussions.

---

[8] See, e.g., https://da.wiktionary.org/wiki/ynkelig.

[9] E.g., https://dsn.dk/soegning/?soegeord=apropos

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
