# OpenReview forum: "Alignment of Wikidata lexemes and Det Centrale Ordregister"
_NoDaLiDa/2023/Conference — NoDaLiDa 2023_

### Official Review · Reviewer_VaY2 · 2023-03-12
**A review of "Alignment of Wikidata lexemes and Det Centrale Ordregister"**

**Rating:** 6
**Confidence:** 3

**Review:**

The paper describes connecting lexemes defined in Det Centrale Ordregister (COR) of Denmark into Wikidata. Even when the two databases represent the same data, their internal representations and capabilities differ. Wikidata lacks many of the feature in COR, maybe due to linguistic features in languages such as Danish not having been considered by Wikidata authors.

The paper describes many of the issues encountered in mapping from COR to Wikidata, discussed in sections for different lexical categories. While the approach is no doubt useful for people familiar with the topic, the paper could be more clear if it attempted to find some structure in the issues described. For example, a table showing the lexical categories and the problems associated with them could make it easier to grasp the big picture. As it is, the paper is currently a list of "this, then this, then this, then this..." and its a bit difficult to see if there is any overarching issue at hand here, or just a list of unconnected problems.

Nevertheless, the paper is clearly intended for people working on this particular domain, and as a description of the issues encountered when attempting the mapping, it should be useful for others working on the same problem.

**Paper Type:**

Short paper

---

### Official Review · Reviewer_Wa93 · 2023-03-16
**Important LR work**

**Rating:** 7
**Confidence:** 3

**Review:**

I appreciate this single-author paper being written in the first person. Appropriate and very good.

The work presents a linking resources between two disparate organisations, which is a major, important, and significant contribution. Connecting these things up is difficult and not necessarily rewarding work. Documenting the process and taking a scientific approach is somewhat unusual and also good.

This non-trivial effort to better represent understanding of Danish strengthens both Wikidata and Det Centrale Ordregister. Beyond that, the scientific aspect of being methodical, of describing that effort, and giving examples, is of intrinsic scholarly value.

This paper is perfect NODALIDA material and I am convinced it would be a mistake not to make place for papers like this at the conf.

**Paper Type:**

Short paper

---

### Decision · Program_Chairs · 2023-03-17

Accept